# Finite Element Analysis of Biomechanical Assessment: Traditional Bilateral Pedicle Screw System vs. Novel Reverse Transdiscal Screw System for Lumbar Degenerative Disc Disease

**DOI:** 10.3390/bioengineering12060671

**Published:** 2025-06-19

**Authors:** Utpal K. Dhar, Kamran Aghayev, Hadi Sultan, Saahas Rajendran, Chi-Tay Tsai, Frank D. Vrionis

**Affiliations:** 1Department of Ocean and Mechanical Engineering, Florida Atlantic University, Boca Raton, FL 33431, USA; udhar2020@fau.edu; 2BHT Clinic, 34307 Istanbul, Türkiye; 3Charles E. Schmidt College of Medicine, Florida Atlantic University, Boca Raton, FL 33431, USA; 4Department of Neurosurgery, Marcus Neuroscience Institute, Boca Raton Regional Hospital, Boca Raton, FL 33486, USA

**Keywords:** disc degeneration, finite element study, range of motion, von Mises stress, lumbar spine

## Abstract

The traditional bilateral pedicle screw system has been used for the treatment of various lumbar spine conditions including advanced degenerative disc disease. However, there is an ongoing need to develop more effective and less invasive techniques. The purpose of this study was to compare the traditional bilateral pedicle screw system (BPSS) with the novel reverse transdiscal screw system (RTSS) for lumbar disc degenerative disease. A 3D solid lumbar L1–L5 spine model was developed and validated based on a human CT scan. Fusions were simulated at L3–L4. The first scenario comprised a transforaminal lumbar interbody cage in combination with the bilateral pedicle screw-rod system (BPSS-TLIF). In the second scenario, the same TLIF cage was combined with reverse L3–L4 transdiscal screws (RTSS-TLIF). Testing parameters included range of motion (ROM) in three orthogonal axes, hardware (cage and screw) stress, and shear load resistance. The ROM of the surgical model was reduced by approximately 90% compared to the intact model at the fused level. The RTSS model demonstrated less ROM compared to the BPSS model at the fused level for all loading conditions. Overall, the RTSS model exhibited lower stress on both screws and cage compared with the BPSS model in all biomechanical testing conditions. The RTSS model also exhibited higher anterior and posterior shear load resistance than the BPSS model. In conclusion, the RTSS model proved superior to the BPSS model in all respects. These findings indicate that the RTSS could serve as a feasible option for patients undergoing lumbar fusion, especially for adjacent segment disease, potentially enhancing surgical outcomes for disc degeneration.

## 1. Introduction

Interbody fusion is a well-established surgical option for a variety of lumbar spine diseases including degenerative, infectious, and traumatic conditions. Standard surgical technique comprises discectomy, insertion of interbody cage, and bone graft into the disc space with a supplementary bilateral pedicle screw–rod stabilization system (BPSS) [1]. It is known that the combination of an interbody cage and a BPSS yields the best fusion rates and, thus, favorable clinical outcomes. Yet the BPSS requires placing two pedicle screws per vertebra and connecting them with rods and is an indirect stabilization technique which relies on moving parts fastened to each other. For a single level stabilization, the required parts are four screws, four tulips, two rods, and four nuts total, making 14 parts [2]. Inherently, such a system is more prone to mechanical failure under physiological multiaxial load. Clinical failure may present as hardware breakage and/or screw loosening. Such long-term complications may lead to pseudoarthrosis and poor clinical outcomes and significantly influence patients’ life quality. There is much need for simpler, stronger, and more resilient hardware options that can yield higher fusion rates, less failures, and better clinical outcomes.

Transdiscal screw (TSS) stabilization is an alternative technique which was developed, assessed, and clinically implemented [3,4]. It comprises only two screws inserted from one vertebra to another through the disc space [5] and is a direct stabilization technique with no moving parts and therefore more resistant to failure. Furthermore, the inherent simplicity of TSS makes it superior to the BPSS in terms of soft tissue damage during implantation. The inferior to superior S1-L5 TSS trajectory was previously developed and implemented [3,4]. This was possible due to the unique anatomical features of the lumbo-sacral segment. Such an infero–superior trajectory is unfeasible for other lumbar segments in the absence of spondylolisthesis due to anatomical restrictions, yet a superior to inferior trajectory was deemed possible. Thus, the model used in this study features a “reverse” transdiscal screw starting at the L3 pedicle and oriented caudally, passing through L3–4-disc space and terminating at the inferior border of the L4 vertebral body. This technique may circumvent the need for exposing and revising existing instrumentation in cases of superior upper adjacent segment disease that typically requires extension of the fusion and reconnection to existing hardware.

Finite element analysis is a biomechanical evaluation tool that provides several advantages in spinal matters, including evaluations of physiological and various pathological and surgical scenarios. It is an alternative method to measure hardware stress which is difficult to obtain from cadaveric studies [6,7,8,9,10,11,12]. In addition, finite element analysis provides reliable simulations of molecular, pathological, and biomechanical characteristics of degenerated discs without the invasiveness and replicability issues associated with other research methods [13,14,15,16,17,18,19]. Studies on the biomechanical efficacy of transdiscal screw approaches for lumbar disc degeneration are still limited. No finite element studies were conducted to compare the traditional BPSS with the RTSS for lumbar disc degeneration. The purpose of this study was to compare biomechanical features of the reverse transdiscal screw system with the standard bilateral pedicle screw–rod system using finite element analysis.

## 2. Materials and Methods

### 2.1. Model Creation

The L1–L5 model was generated using the computed tomography (CT) imaging data of a healthy patient (64 years old, female) in Mimics 25.0 (Materialize, Leuven, Belgium). The volunteer was informed; consent forms authorizing the use of her images for educational and research purposes were secured. All methods were carried out in accordance with relevant guidelines and regulations. All experimental protocols were approved by Boca Raton Regional Hospital and Florida Atlantic University. Figure 1 shows how a typical CT scan is converted into a 3D model through Mimics software. Digital Imaging and Medical Communication (DICOM) imaging data of the patient’s spine are transferred into Mimics. Based on these images, generation of vertebral bodies was achieved through threshold segmentation, yielding surface mesh elements that are exported through STL format into 3-Matic (Materialize, Leuven, Belgium) for further operations. Careful post-processing in 3-Matic was then carried out for geometry mesh preparation. Following this, the cortical and cancellous bones were derived from the processed vertebral body with an offset of 1 mm [20]. The endplates were generated with a thickness of 0.5 mm on the top and bottom surfaces of the cortical bone, with generation of the disc and its constituent components, comprising the annulus fibrosis and nucleus pulposus, connecting contiguous endplates within the same software. Meshing of cortical and cancellous bodies was then finally completed using tetrahedral elements.

A Unilateral L3–L4 facetectomy was performed. After excision of the posterior longitudinal ligament and annulus fibrosus, the nucleus pulposus was removed. An appropriately sized TLIF cage was designed with CAD software (SolidWorks 2023, Dassault Systèmes Inc., Vélizy-Villacoubla, France) based on the L3–L4 gap and was placed. In the BPSS model, four pedicle screws and two rods were also crafted with the same CAD software and employed to fix L3 and L4. Figure 2A–C depict the BPSS model from sagittal, anterior, and posterior perspectives, respectively. In the RTSS model, a similar TLIF cage was inserted between L3 and L4. Two transdiscal screws passed through the L3 pedicle inferiorly and terminated at the inferior border of the L4 vertebral body. The RTSS model is displayed in sagittal, anterior, and posterior views in Figure 2D–F, respectively. In both the BPSS and RTSS models, the diameter of the screw size was chosen to be 5.5 mm. The length of the transdiscal screw was 68 mm, while the pedicle screw length was 40 mm.

### 2.2. Finite Element Model

The completed components were then exported. In Ansys Workbench (Ansys Workbench 2023 R1, Ansys Inc. Canonsburg, PA, USA), the annulus geometry prepared in 3-Matic was meshed using a hexahedral mesh with embedded tension-only fiber bar elements for simulation of the ground substance and fibrous components, respectively. Fibers were constructed in a crisscross pattern with angles of approximately 30° to the horizontal and Young moduli of 550 MPa in the outside band to 360 MPa in the inner band. The intermediate layers varied linearly in their Young moduli. Detailed material properties are listed in Table 1 for the constructed model with the corresponding literature. Ligaments were subsequently added to the model in Ansys Workbench and programmed as tension-only spring elements. Based on prior studies, nonlinear force–displacement relationships, which describe the response of each ligament to varying vertebral loads, were also incorporated. Similarly, ligament material properties were referenced in Table 1. The included ligaments in this model consisted of the anterior longitudinal ligament, posterior longitudinal ligament, transverse ligaments, spinous and supraspinous ligaments, and capsular ligaments.

The mesh convergence was performed for the intact model. Three distinct meshes such as Mesh1, Mesh2, and Mesh3 were created consecutively for different parts of the bone. Mesh1 has the lowest number of elements and nodes, whereas Mesh3 has the highest among the three meshes. For the endplate and nucleus pulposus, a relatively fine mesh was chosen. The difference between Mesh1 and Mesh2 showed more than 10%, whereas it was less than 5% between Mesh2 and Mesh3 when accounting for equivalent von Mises stresses for the identical loading conditions. The mesh was deemed to converge when the variance between the predicted von Mises stresses of various components from two consecutive mesh refinements was below 5% under a torque of 7.5 Nm. Hence, Mesh2 was chosen for the entire FE analysis. The final model consisted of 435,808 elements and 787,522 nodes, comprising five vertebrae with their corresponding interconnected discs, ligaments, and connections.

### 2.3. Contact, Loading, and Boundary Conditions

Model loading and boundary conditions were applied in accordance with cadaveric experiments previously described [21,22]. For the simulations, the inferior surface of the L5 vertebral body was fixed and a 7.5 N.m. moment was applied to the upper endplate of L1. This moment was varied in direction to simulate each of the loading conditions. Furthermore, in each load case, 500 N of compressive force was applied and evenly distributed across the upper endplate of L1.

**Table 1 bioengineering-12-00671-t001:** Element type and material properties used in the model.

Component	Young Modulus (Mpa)	Poisson Ratio	Element Type
Cortical [23]	12,000	0.3	Tetrahedral element
Cancellous [23,24]	100	0.2	Tetrahedral element
Endplate [25]	500	0.45	Tetrahedral element
Annulus ground [23]	4.2	0.45	Hexahedral element
Annulus fiber [17,26,27,28]	360–450	Cross-sectional area (0.15 mm^2^)	Link element
Ligament [23,28,29,30]	Calibrated force–deflection curve		Spring element
Nucleus [29]	1	0.4	Tetrahedral element
Titanium [30]	110,000	0.3	Tetrahedral element

Lastly, anterior and posterior shear loads of 150 N were applied parallel to the upper endplate of the L3 by fixing the bottom surface of L4. A representative boundary condition for the intact model is illustrated in Figure 3A for axial rotation and Figure 3B for anterior shear loading. Because of the body-dependent mesh sizing, which would lead to highly distorted elements for shared nodes between bodies, bonded contacts were utilized to connect the vertebral discs to the endplates. Bonded contacts were also utilized for connection of the endplates to the vertebral bodies. All implants were similarly connected to the existing vertebral bodies using bonded connections, with Boolean operations where appropriate to achieve a cohesive model.

## 3. Results

### 3.1. FE Model Validation

After subjecting similar boundary and loading conditions, we compared our range of motion (ROM) with those derived from a cadaveric investigation conducted by previous in vitro experiment and other FEA study [31,32]. Figure 4 illustrates the ROM of the current study compared with the experimental and FEA results and congruence with the previously documented data.

Consequently, the intact model in the present study was successfully established and was deemed ready for further analysis.

### 3.2. ROM

In Figure 5, the ROM for flexion, extension, left bending, right bending, left rotation, and right rotation is compared between the BPSS and RTSS models for the index level. The ROM of the BPSS and the RTSS was compared with the intact lumbar model. Both treatment models showed significant reduction in ROM across all loading scenarios when compared to the intact model. Additionally, the RTSS model demonstrated a further reduction in ROM compared to the BPSS model under all loading conditions. The greatest difference between the two models was observed in extension, where the RTSS model reduced ROM by 58% compared to the BPSS model. The smallest difference was seen in right rotation, with a 13% reduction.

### 3.3. Cage von Mises Stress

The cage stress under various loading scenarios was calculated for both the BPSS and the RTSS models and is shown in Figure 6b, while Figure 6a is the typical contour plot for cage von Mises stress for the RTSS model in flexion condition. In both flexion and extension, the cage stress in the BPSS model was over 30% higher compared to the RTSS model. For rotation and right bending, there was no significant difference in cage stress between the two models. In left bending, the BPSS model exhibited a cage stress of 86.5 MPa, while the RTSS model showed a significantly lower stress of 49.6 MPa. The highest stress for the BPSS system was observed in flexion, reaching 88.7 MPa, whereas for the RTSS system, the highest stress occurred in left rotation, measuring 67.3 MPa.

### 3.4. Screw von Mises Stress

A typical von Mises stress for the BPSS and RTSS models is shown in Figure 7a and Figure 7b, respectively, for the flexion condition. Figure 7c illustrates the screw stress for the BPSS and the RTSS models under various physiological loading conditions. In the BPSS model, higher stress was observed during axial rotation compared to flexion–extension and lateral bending scenarios. The maximum screw stress in the BPSS model was recorded at 418 MPa during right rotation, while the minimum stress was 208 MPa during right bending. In contrast, the RTSS model exhibited the highest screw stress (239 MPa) during extension and the lowest stress (108 MPa) during left bending. In lateral bending, the RTSS model showed a 40% lower screw stress compared to the BPSS model. For extension, the difference in screw stress between the RTSS and BPSS models was minimal, with both recording a value of 239 MPa, a smaller variation compared to other loading conditions.

### 3.5. Resistance to Shear Load

Regarding anterior and posterior shear loads, a shear load of magnitude 150 N was applied at the center of the L3 vertebra in a horizontal direction parallel to the base of L4. These loads replicated the loading scenarios in the mechanical assessment of cadaveric motion segments [33]. The displacement comparison due to shear load is depicted in Figure 8. The RTSS model illustrated anterior and posterior displacement of 3.43 mm and 3.44 mm, respectively, whereas the BPSS model showed 3.70 mm and 4.48 mm. No notable difference was experienced in these two models due to the anterior shear load, but the posterior shear load RTSS model exhibited 23% lower displacement than the BPSS model. Overall, the RTSS model showed better shear load resistance than the BPSS model for the same loading condition.

## 4. Discussion

In the present study, we compared the RTSS and BPSS models to evaluate their mechanical properties. This study focuses on a novel RTSS designed to address some of the biomechanical shortcomings associated with the traditional BPSS, particularly reducing the incidence of non-union and improving post-surgical outcomes. In line with the previous studies [3,4], the RTSS demonstrated higher stiffness across all loading conditions, particularly in extension, where a 58% reduction in ROM was observed compared to the BPSS model. These findings suggest that the RTSS provides enhanced rigidity, which could potentially facilitate fusion.

The stress analysis of both the cage and screws shows the advantages of the RTSS. The cage stress was notably lower in the RTSS model under flexion and lateral bending conditions, reduced by over 30% compared to the BPSS. These results indicate that the RTSS may distribute mechanical loads more evenly, avoiding high pressure points, and can potentially reduce the risk of cage-related complications such as subsidence and migration [34]. Similarly, screw stress was significantly lower in the RTSS model during lateral bending and axial rotation, with a 40% reduction compared to the BPSS model. This decrease in stress is crucial for long-term stability, as reduced screw stress may lower the risk of hardware breakage or loosening—potential complications in spinal fusion surgeries [35]. The screw stress ranged from 200 MPa to 700 MPa, while the cage stress was approximately 100 MPa when this type of construction is inserted in the lumbar spine. In this study, we found that the von Mises stresses were significantly lower than the yield strength of titanium, indicating that this construction can be safely used in real scenarios.

The vertebral slippage was investigated after applying the anterior and posterior shear force. Most biomechanical studies have neglected this parameter in their analysis. This information is necessary for daily activities that produce such loads, including bending and lifting [36]. Overall, the magnitude of displacement by posterior shear load was lower than the anterior shear force condition for both the BPSS and RTSS models. Lee et al. [33] also reported decreases in displacement of the upper vertebra due to posterior shear loading relative to anterior shear loading. In this study, the RTSS model exhibited higher resistance to both anterior and posterior shear loading compared to the BPSS model, further highlighting the advantage of the RTSS. We attribute this effect to the geometry of the screw in relation to the axis of the spine. The RTSS axis is almost parallel to the spine, whereas the BPSS screws are orthogonal to the spine’s axis. Thus, the RTSS is much more resistant to orthogonal forces (shear force) in comparison to the BPSS. Overall increased stiffness in combination with lesser hardware stress theoretically makes the RTSS a much more effective alternative to the BPSS. Moreover, it is potentially useful in adjacent segment disease where typically the surgeon needs to expose and connect with the previous instrumentation. But reverse TSS technique bypasses the need. The RTSS can be less invasive than the BPSS even with modern paramedian muscle splitting minimally invasive techniques. As the RTSS does not require a tulip for the screw head, it can be placed entirely percutaneously through a 1 cm incision on each side. Biplanar fluoroscopy or an image-guided technique is clearly needed for such placement.

In the mechanical perspective view, the RTSS has larger bone–interface contact and a longer lever arm. This scenario will enhance the pull-out strength and resistance to shear load. Another advantage of the RTSS is its insertion angle of approximately 45 degrees, compared to the BPSS, where screws are typically inserted parallel to the horizontal plane. Due to this oblique trajectory, the resulting compressive force acting on the RTSS is reduced to approximately two-thirds of that experienced by the BPSS. That could be another reason the RTSS screws experience lower stresses than the BPSS. That could potentially lower the risk of implant failure under axial loading. In the RTSS, the screws align more directly with the load-bearing axis of the spinal column. This allows for more axial load sharing between the bone and implant. On the other hand, in the BPSS, screws are inserted to the posterior side and somewhat off-axis, which is responsible for higher bending moments at the screw–rod junction. In the osteoporotic bone, the RTSS may offer greater initial fixation and long-term stability because of the load distribution along the transdiscal path.

While these biomechanical advantages highlight the potential benefits of the RTSS, it is important to consider the limitations of FEA. Only clinical studies can fully unlock the potential of new stabilization techniques. FEA studies of spinal implants rely on a single spine model and may not be able to fully account for clinical pathologies including disc degeneration, limiting the clinical application of the results to a wide population. Additionally, some models assume linear material properties and ignore the contributions of paravertebral muscles which may not be fully accurate when considering vivo loading conditions.

To further improve the RTSS design and optimization, it is necessary to analyze the multiple insertion angle by using FE analyses. Screws could be designed as double thread to improve the anchorage in both cortical and cancellous bone. Diameter and length could be customized to reduce the damage of the endplate. Moreover, interbacterial coating on the screws could be useful for reducing post-operative infection. Hydroxyapatite or titanium plasma spray could be used for promoting osteointegration.

In finite element modeling, the material assignment for different components is very important, as the simulation influences outcome significantly. Therefore, careful selection of appropriate material properties is essential for obtaining meaningful and reliable results. In this study, we also employed an appropriate boundary condition as they play a critical role in accurately replicating physiological loading scenarios. Moreover, mesh sensitivity was performed to check the mesh quality. It is essential to validate the current model by comparing it with existing experimental and other finite element studies to ensure the results are accurate and meaningful. Common mistakes should be avoided such as misaligning the anatomical landmark, over constraining the model that may affect the outcome significantly. This insight will assist other engineers not only with this type of disc degeneration model but with common biomechanical design projects.

## 5. Conclusions

These findings suggest that the RTSS may offer a viable alternative for patients undergoing lumbar fusion, potentially improving overall surgical outcomes for disc degeneration. However, further research, including in vivo studies and long-term clinical trials, is necessary to fully assess the clinical impact of the RTSS and its potential role in the treatment of lumbar degenerative disc disease.

## 6. Limitations

This BPSS compared with the RTSS for lumbar disc degeneration is grounded in FE techniques and presents specific limitations. First, we utilized skeletal data from only one individual’s FEA simulation, without considering individual differences. Secondly, applying a compressive force of 500 N combined with a moment of 7.5 Nm does not precisely simulate the stress experienced by the cage and screws in patients following spinal fusion surgery. Another limitation of this study is that the RTSS may be technically demanding to be inserted in such a position, and it would not be suitable for the multi-level fixation. Finally, in the present model, ligaments are modeled as nonlinear spring elements that are affected only by tensile forces, while muscles are excluded from the simulation. This may influence the movement and stress variations in the lumbar spine after post-surgery.

## Figures and Tables

**Figure 1 bioengineering-12-00671-f001:**
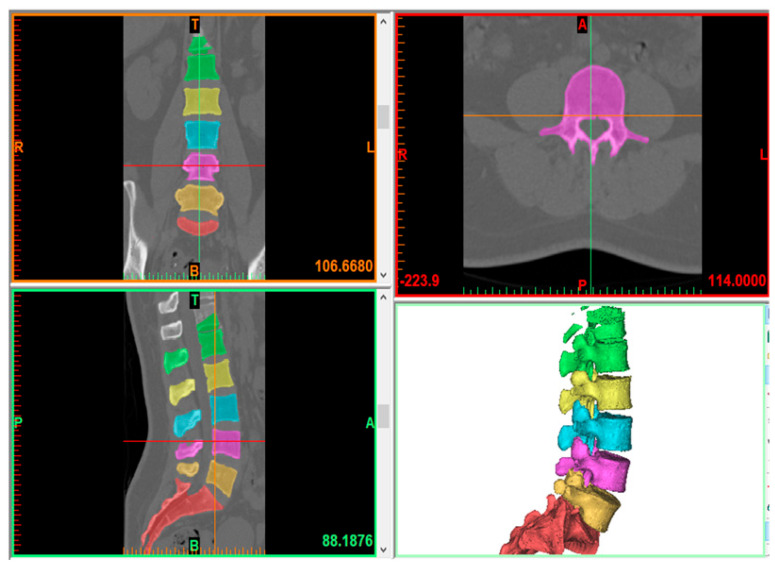
A typical diagram for converting a 2D CT scan to a 3D model.

**Figure 2 bioengineering-12-00671-f002:**
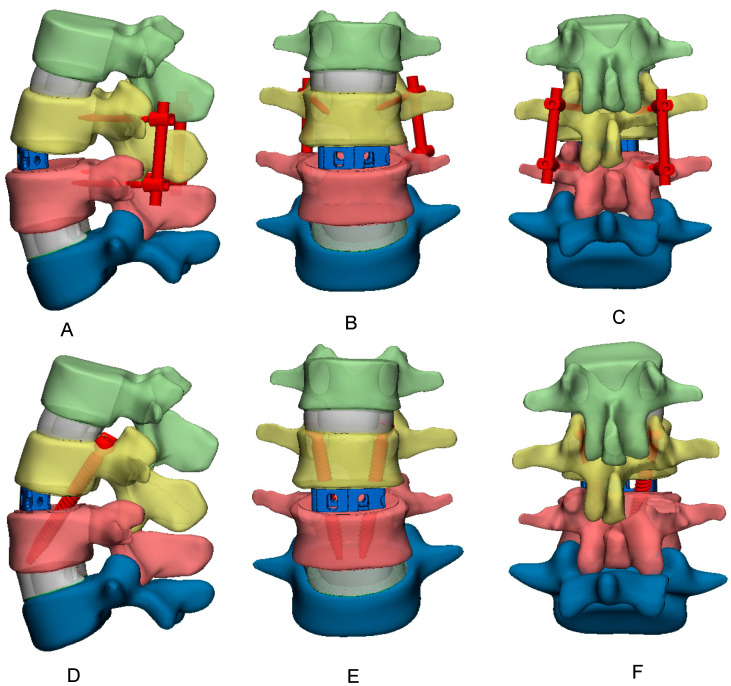
Traditional bilateral pedicle screw system: sagittal (**A**), anterior (**B**), and posterior view (**C**); reverse trandiscal screw system: sagittal (**D**), anterior (**E**), and posterior view (**F**).

**Figure 3 bioengineering-12-00671-f003:**
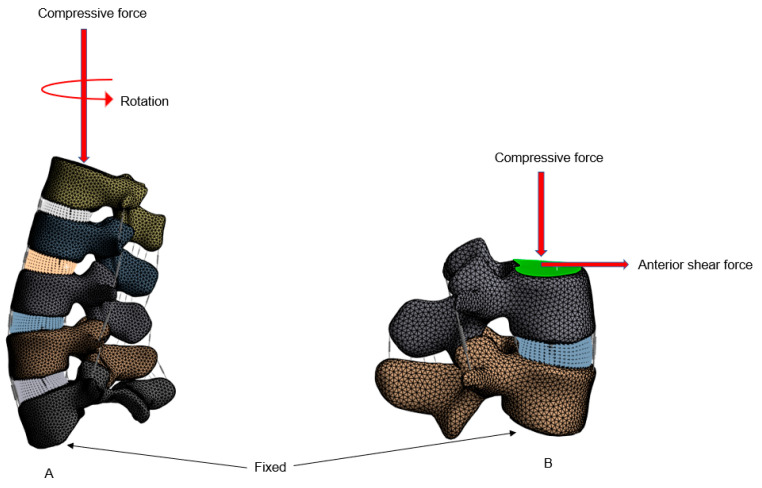
A typical boundary condition for left-axial rotation (**A**) and anterior shear loading (**B**) for the intact model.

**Figure 4 bioengineering-12-00671-f004:**
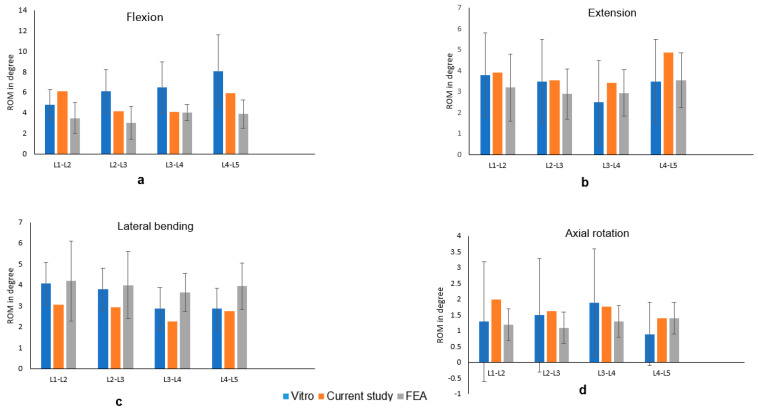
Comparison of ROM with the in vitro and FEA study for flexion (**a**), extension (**b**), lateral bending (**c**), axial rotation (**d**).

**Figure 5 bioengineering-12-00671-f005:**
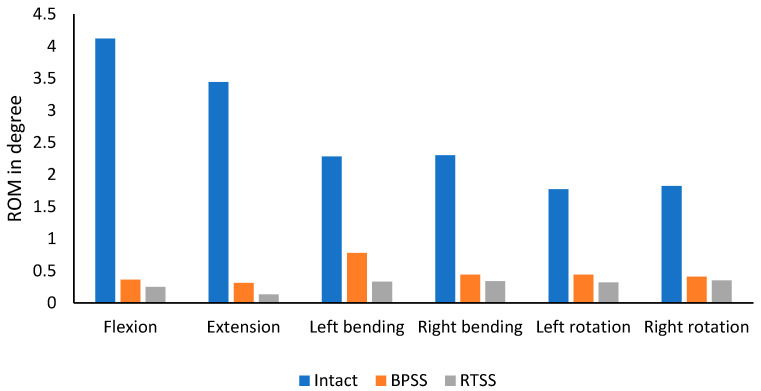
Comparing range of motion between the intact, BPSS, and RTSS models.

**Figure 6 bioengineering-12-00671-f006:**
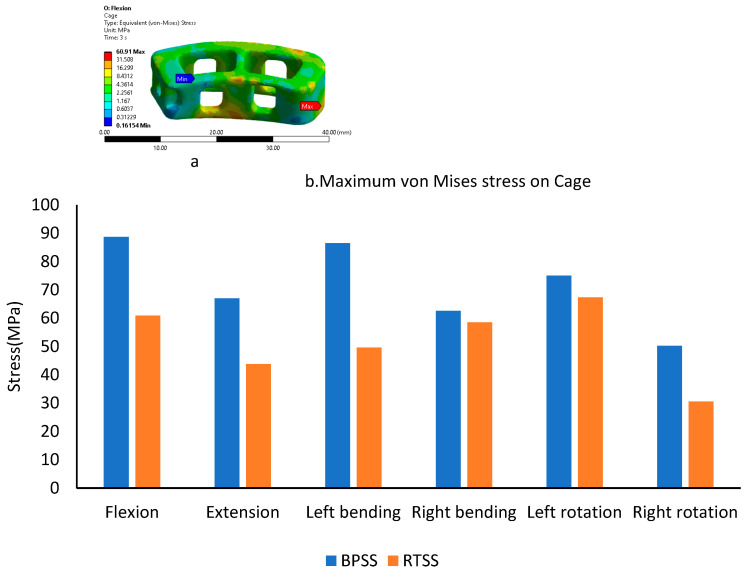
A typical von Mises stress contour on RTSS cage under flexion condition (**a**). Comparing maximum von Mises stress on the cage between the BPSS and the RTSS (**b**).

**Figure 7 bioengineering-12-00671-f007:**
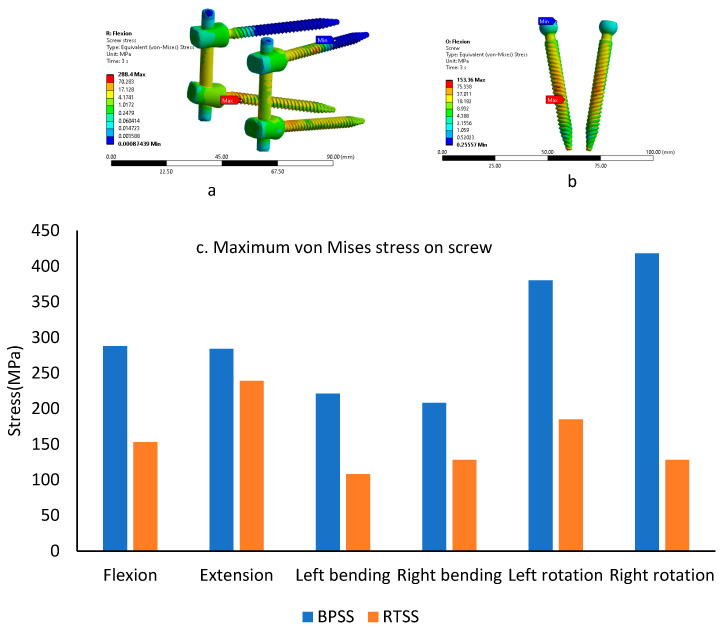
A typical von Mises stress contour on internal fixation under flexion condition (**a**). BPSS (**b**). RTSS (**c**). Comparing maximum von Mises stress on the screw between the BPSS and the RTSS.

**Figure 8 bioengineering-12-00671-f008:**
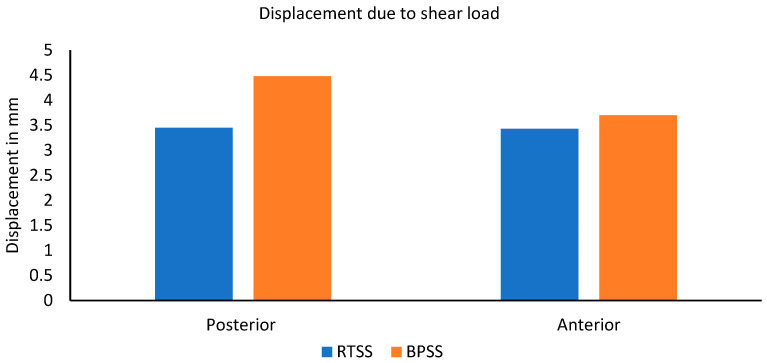
Comparing shear load resistance between the RTSS and BPSS models due to anterior and posterior shear load.

## Data Availability

The datasets used and/or analyzed during the current study are available from the corresponding author on reasonable request.

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
