# Peer review of "Finite Element Analysis of Biomechanical Assessment: Traditional Bilateral Pedicle Screw System vs. Novel Reverse Transdiscal Screw System for Lumbar Degenerative Disc Disease"

_bioengineering, 2025, doi:10.3390/bioengineering12060671_

Round 1

Reviewer 1 Report

Comments and Suggestions for Authors

This manuscript concerns a case study where a conventional screw system is compared with a new design for lumbar degenerative disc disease, by means of stress analysis using finite element modelling. Although an important disease, the manuscript appears to be like a technical report using classical techniques for a specific and limited situation. Of more general interest is the methodology, and a wider readership, beyond specialists for the given case, could be reached if this work would more clearly highlight the methodology, and how it could be applied to other design situations. Both in abstract, introduction and discussions, the authors could showcase lessons learned, mistakes to be avoided to help other engineers and researchers not only working on this type of lumbar disc support. It would be useful to indicate target, limit or desired values in the design space in the bar charts, to see where the aim is. A section on further design improvements, towards optimization would also be useful. Just because a better solution has been shown, does not mean that it could be significantly further improved.

Otherwise, I find the manuscript well written, although some improvements could be done font size in figures. The references are not written in uniform style.

Reviewer 2 Report

Comments and Suggestions for Authors

This manuscript compared the BPSS and RTSS with FEA, the ROM and stresses in the screws are comprehensively inspected. Overall, the manuscript is well organized, while the “Discussion” sections should be enhanced and deep for the “mechanical” perspective. For example, how these two screw systems work, in terms of load sharing or different stress distribution patterns? Current manuscript just a report like, and the depth is lacking. Please spend more effort on this.

In addition, minor but annoying, the figures are very rough and not professional, I have listed details suggestions below:

Figure 1 is not necessary. Also, it should be “CT scan”, rather than “Ctscan” in the legend.

Figure 2D, E, and F are not clearly show the RTSS, adjusting the transparency or using a sectional view to expose the screw would be better.

Figure 3B, the arrow is missing for the compressive force, while the Figure3A has it.

Figure 4 the legend is not precise, especially “vitro”, “Current study”, and “FEA” should be well defined, current work also use FEA, right? Should be “Vitro”, rather than “vitro”

Figure 5 there should be a “,” after BPSS

Figures 2, 3, 6, and 7, pictures from FEA software, a white background is suggested.

One more suggestion on Figures, font size is not consistent, font color should be black but now gray.

Round 2

Reviewer 2 Report

Comments and Suggestions for Authors

My comments have been addressed well. The Figures still can be improved, For example, the axia of the bar chart in Figure 4, 5, 6, and 7 can be black color rather than gray, the y axis is missing. 

Author Response

Reviewer2

My comments have been addressed well. The Figures still can be improved, For example, the axia of the bar chart in Figure 4, 5, 6, and 7 can be black color rather than gray, the y axis is missing. 

Response: 

Thank you for your valuable feedback. We have updated Figures 4, 5, 6, 7, and 8 by changing all axis colors from gray to black. Additionally, we have restored the missing y-axis. We believe the figures now appear more attractive and visually appealing.

Thank you once again for helping us improve the quality of our figures.